# The Role of Estrogen Receptors in Cardiovascular Disease

**DOI:** 10.3390/ijms21124314

**Published:** 2020-06-17

**Authors:** Laila Aryan, David Younessi, Michael Zargari, Somanshu Banerjee, Jacqueline Agopian, Shadie Rahman, Reza Borna, Gregoire Ruffenach, Soban Umar, Mansoureh Eghbali

**Affiliations:** Department of Anesthesiology, Division of Molecular Medicine, David Geffen School of Medicine at University of California, Los Angeles, BH-550 CHS, Los Angeles, CA 90095-7115, USA; laryan@ucla.edu (L.A.); davidyounessi21@gmail.com (D.Y.); mikezargari@ucla.edu (M.Z.); SoBanerjee@mednet.ucla.edu (S.B.); JAgopian@mednet.ucla.edu (J.A.); Shadierahman@gmail.com (S.R.); RBorna@mednet.ucla.edu (R.B.); GRuffenach@mednet.ucla.edu (G.R.); SUmar@mednet.ucla.edu (S.U.)

**Keywords:** estrogen, estrogen receptors, estrogen receptor alpha, estrogen receptor beta, GPR30, hypertension, atherosclerosis, ischemia-reperfusion injury, heart failure with reduced ejection fraction, heart failure with preserved ejection fraction

## Abstract

Cardiovascular Diseases (CVDs) are the leading cause of death globally. More than 17 million people die worldwide from CVD per year. There is considerable evidence suggesting that estrogen modulates cardiovascular physiology and function in both health and disease, and that it could potentially serve as a cardioprotective agent. The effects of estrogen on cardiovascular function are mediated by nuclear and membrane estrogen receptors (ERs), including estrogen receptor alpha (ERα), estrogen receptor beta (ERβ), and G-protein-coupled ER (GPR30 or GPER). Receptor binding in turn confers pleiotropic effects through both genomic and non-genomic signaling to maintain cardiovascular homeostasis. Each ER has been implicated in multiple pre-clinical cardiovascular disease models. This review will discuss current reports on the underlying molecular mechanisms of the ERs in regulating vascular pathology, with a special emphasis on hypertension, pulmonary hypertension, and atherosclerosis, as well as in regulating cardiac pathology, with a particular emphasis on ischemia/reperfusion injury, heart failure with reduced ejection fraction, and heart failure with preserved ejection fraction.

## 1. Introduction

Although cardiovascular disease (CVD) remains the primary cause of death in both men and women, epidemiological studies indicate that premenopausal women are more protected against the development of CVD compared to age-matched men [1]. Such advantages have alluded to the involvement of sex hormones in mediating the cardioprotective effects in premenopausal women [2]. 17beta-estradiol (E2) is the most common form of circulating estrogen as well as the major female sex hormone. As such, most of the data regarding estrogen signaling refers to E2. Indeed, E2 levels are inversely associated with CVD events in post-menopausal women [3,4]. Thus, E2 signaling is believed to play a significant role in CVD pathophysiology.

E2 exerts its effects through both genomic and non-genomic actions [5]. E2 binds to the classical estrogen receptors (ERs), estrogen receptor alpha (ERα) and estrogen receptor beta (ERβ) in the cytosol. The E2-bound ER complexes undergo conformational changes to dimerize, translocate to the nucleus, and either directly bind to DNA sequences known as estrogen response elements (EREs) or indirectly bind to DNA through other transcription factors to differentially regulate gene transcription [5,6]. Traditionally, E2 has been known to exert its cardioprotective effects by binding to the nuclear receptors ERα and ERβ. However, G protein–coupled receptor GPR30 (G protein–coupled estrogen receptor 1 or GPER) has also gained increased research attention over the past decade. GPR30 is localized in the endoplasmic reticulum and plasma membrane, and is known to be expressed in cardiomyocytes [7,8,9]. E2 binds to GPR30 to exert rapid non-genomic events, triggering intracellular signaling cascades that alter gene expression downstream [4,10]. A wide array of studies have highlighted several beneficial effects of E2 treatment on the cardiovascular system [2,11]. Such effects are associated with reduced fibrosis, reduced oxidative stress, improved mitochondrial function, attenuation of cardiac hypertrophy, and stimulation of angiogenesis and vasodilation [3,11]. Although a significant number of studies have been published on the biological effects of E2 action in cardiovascular function, the selectivity of each ER in the regulation of the E2-mediated effects remains poorly understood. Thus, understanding the mechanisms of action for E2 is important in order to devise more effective therapeutic strategies to prevent cardiovascular events. 

The specific function of ERα, ERβ, and GPR30 and the mechanisms by which they confer cardioprotection is studied actively in various pre-clinical animal models. This review will discuss current reports on the underlying molecular mechanisms of the ERs in regulating vascular function, with a special emphasis on hypertension, pulmonary hypertension, atherosclerosis, as well as in regulating cardiac function, with a particular emphasis on ischemia/reperfusion injury, heart failure with reduced ejection fraction (HFrEF), and heart failure with preserved ejection fraction (HFpEF). 

## 2. Role of Estrogen Receptors in Vascular Pathology

### 2.1. Hypertension

According to the American Heart Association, hypertension (HTN) is characterized by a chronic elevation of systemic arterial pressure above a certain threshold value (≥130 mm Hg systolic or ≥80 mm Hg diastolic) [12]. HTN is the most significant modifiable risk factor for CVD, with an estimated half of CVD events attributed to it [13,14]. The incidence and prevalence of HTN differs among men and women [15]. According to the Centers for Disease Control and Prevention, under the age of 60, the prevalence of HTN is higher in men; however, over the age of 60, HTN is more common in women [16]. Sex steroids are found to be the key players behind the sex-related differences in HTN [17]. E2 is known to have powerful acute and chronic vasodilator activity, ultimately leading to lowering blood pressure (BP) [18]. The majority of animal studies suggest that E2 engages in numerous mechanisms that guard against HTN, such as the stimulation of the nitric oxide (NO)-mediated vasodilator pathway [19,20]. In this section of the review, we have described the actions of E2 in BP regulation and HTN management, with special emphasis on its genomic and rapid non-genomic actions mediated by cytosolic, nuclear, and membrane-bound ERs respectively. 

ERα has been shown to mediate the protective effects of E2 in Angiotensin II (Ang II)-induced HTN in female mice [21]. Here, Ang II-induced HTN was increased in ovariectomized (OVX) ERα knockout (KO) female mice compared to intact controls, which involved an increase in sympathetic outflow, suggesting that E2 acts primarily through ERα in mediating the protective effects against AngII-induced HTN. A more recent study conducted on female mice expands on this finding suggesting the protective role of ERα against AngII-induced HTN is dependent on ERα gene transcription through activation function 2 (AF2) and not membrane ERα signaling [22]. Furthermore, administration of a highly selective ERα agonist, Cpd1471, has been shown to prevent endothelial dysfunction in OVX spontaneously hypertensive rats. Here, expression levels of aortic endothelial nitric oxide synthase (eNOS) were reduced in OVX spontaneously hypertensive rats and normalized upon administration of both E2 and Cpd1471, validating the E2-mediated protective effects of ERα against HTN [23]. 

Studies have delved into the physiological function and protective role of ERβ against HTN [24,25]. E2 has been shown to attenuate vasoconstriction in mice via an ERβ-mediated increase in inducible NOS expression (iNOS) [24]. In contrast, ERB-deficient male mice develop abnormal vascular contraction, ion channel dysfunction, and HTN, and exhibit augmented vasoconstriction in their blood vessels. Interestingly, E2 treatment of cells transfected with ERα resulted in suppression of iNOS activity, suggesting the antagonizing effect of ERα in the ERβ-mediated induction of iNOS. Furthermore, selective ERβ agonist 8β-VE2 administration to OVX spontaneously hypertensive rats attenuated vascular resistance and HTN, primarily by a reduction in systolic BP [25]. 

GPR30 also plays a pivotal role in inducing the anti-hypertensive effects of E2 [18]. Selective activation of GPR30 lowers BP through rapid vasodilation and protects the heart from hypertensive injury [26]. During HTN, the rate of proliferation and migration of vascular smooth muscle cells (VSMC) increases [27]. Interestingly, E2 has the potential to decrease the VSMC proliferation rate after injury [28,29]. However, the anti-proliferative actions of E2 are not mediated through ERα or ERβ, as E2 maintains its anti-proliferative effect in both ERα and ERβ KO mice, indicating that the E2 response could be mediated by other receptors such as GPR30 [10,29]. Activation of GPR30 with G1, a highly specific GPR30 agonist, has been shown to decrease coronary VSMC proliferation and migration [10]. Furthermore, the activation of GPR30 by G1 is known to phosphorylate (activate) eNOS and subsequently, induce NO-mediated vasodilation in human endothelial [30]. The G1 activation of NO formation is shown to be mediated by various pathways such as proto-oncogene tyrosine-protein kinase (c-Src), epidermal growth factor receptor (EGFR), phosphoinositide-3-kinase (PI3K), and extracellular-signal-regulated kinase (ERK) [30]. Interestingly, these pathways have been shown to be activated during pregnancy, when there is a rapid increase in E2 levels [31,32]. The protective effects of GPR30 activation were further validated by pharmacological pretreatment of telomerase-immortalized human umbilical vein endothelial cells with GPR30-selective antagonist, G36, which reduced NO production in response to E2 and impeded the vasodilatory effects of G1 [30]. 

Taken together, it is yet unknown which ER is most responsible for mediating the protective effects of estrogen against hypertension, as previous studies have shown all three ERs to play a protective role to some extent. ERα activation has been shown to reduce endothelial dysfunction. ERβ activation decreases BP, vasonconstriction, vascular resistance, and attenuates cardiac hypertrophy. Finally, GPR30 reduces BP, stimulates vasodilation, and decreases VSMC proliferation and migration (Table 1, Figure 1).

### 2.2. Pulmonary Hypertension

Pulmonary arterial hypertension (PAH) is characterized by pulmonary vascular remodeling resulting in right ventricular (RV) hypertrophy and eventually, severe decompensated RV failure and death [33]. Almost four decades ago, Rabinovitch et al. demonstrated that female rats develop less severe pulmonary hypertension (PH) compared to males [34]. Although the incidence of PH is more prevalent in females, various animal studies have mainly demonstrated that females develop less severe PH compared to males—a phenomenon known as the “estrogen paradox” [34,35,36]. E2 pre-treatment has been shown to reduce PH severity in both female [37] and male rats [38]. Since PH is not always diagnosed early, Umar et al. explored the ability of E2 to rescue preexisting PH [39]. They found that E2 delivery, even after establishment of severe PH, can restore lung and RV structure and function in male rats—a protective effect that is maintained even after E2 removal. Higher E2 levels are also associated with improved RV systolic function in postmenopausal females on hormone replacement therapy, and female PAH patients have a higher baseline RV ejection fraction (RVEF) response after medical intervention compared to males—further cementing the protective role of E2 in women [40,41]. To better understand this relationship, a thorough understanding of implicated ERs is paramount. 

In its current state of understanding, the role of ERα in PAH remain controversial [42,43]. Frump et al. measured the expression of ERα, ERβ, and GPR30 in the RV (a major determinant of survival in PAH) of rats with SU5416/hypoxia-induced PH (Su/Hx PH) [42]. ERα expression is found to decrease in the RV of both intact (non-significant) and OVX (significant) Su/Hx female rats. Therapeutic E2 introduction showed restoration of ERα expression levels in OVX Su/Hx rats to that of untreated controls, and a negative correlation was identified between RV ERα expression and both RV systolic pressure as well as RV hypertrophy. ERβ and GPR30 expression were also measured, but no significant relationship was identified. Finally, treatment with ER-specific agonists to male rats confirmed that the ERα-specific agonist is the most consistent and potent in mediating the effect of E2 on the RV—further supporting the protective role of ERα against PH [42]. Another proposed relationship between ERα and PAH relies on differential expression of bone morphogenetic protein receptor type 2 (BMPR2) [44]. BMPR2 is a receptor for transforming growth factor (TGF)-β and bone morphogenetic protein, both of which play a role in the growth response of pulmonary artery smooth muscle cells (PASMC) [45]. Mutation of the BMPR2 gene can give rise to heritable PAH and abnormally low expression of BMPR2 can lead to PAH predisposition [46,47]. When analyzing BMPR2 expression in both in vitro and in vivo studies, BMPR2 is found to be expressed at significantly higher levels in male mice lungs compared to OVX females. Considering that ERα is known to downregulate BMPR2 expression through a promoter-binding site, this disparity can be explained by increased expression of ERα among females [44]. 

Interestingly, it has been shown that female (but not male) mice overexpressing serotonin transporter (SERT+) develop severe PH, which is abolished by ovariectomy [48]. Here, chronic administration of E2 was shown to reestablish PH in OVX SERT+ mice. The same group also reported that serotonin increases the expression of ERα [43]. They show that ERα antagonist MPP reduces the severity of PH by decreasing pulmonary vascular remodeling, RV systolic pressure, and PASMC proliferation. Conversely, ERα agonist PPT has a proliferative effect on human PASMC in vitro. It is suggested that ERα mediates the estrogen-induced proliferation of human PASMC via mitogen-activated protein kinase (MAPK) and protein kinase B (Akt) signaling [43]. Lastly, ERα expression is higher in PASMC isolated from female PAH patients compared to male patients [43]. In summary, though there is evidence in support of the protective role of E2 and ERα against PH, the aforementioned controversial findings suggest a need for further research. 

ERβ is known to be a viable cardiopulmonary protective receptor whose activation elicits an antiproliferative and antifibrotic response [49,50]. Studies have shown ERβ to play a crucial role in mitigating the severity and damage of PH [51,52]. Notably, Umar et al. reported that introduction of ERβ antagonist PHTPP to male rats prevents the rescue function of E2, and that an ERβ-specific agonist is as effective as E2 in PH rescue [39]. Here, they concluded that E2 rescue of PH is likely mediated through ERβ, and that this rescue involves the suppression of fibrosis, inflammation, and RV hypertrophy. Frump et al. further explored the impact of E2 treatment on the progression of hypoxia-induced PH in Sprague-Dawley male rats and human pulmonary artery endothelial cells [49]. To determine the contribution of each ER to pulmonary vascular remodeling, they measured the expression of ERα and ERβ mRNA in hypoxic conditions and reported that ERβ is transcriptionally upregulated in both pulmonary vasculature and isolated pulmonary artery endothelial cells. To confirm their hypothesis that E2′s therapeutic effects act through a hypoxia-inducible factor 1α (HIF-1α)-dependent increase in ERβ expression, they delivered deferoxamine, a HIF-1α stabilizer, and a HIF-1α knockdown to rat pulmonary artery endothelial cells. The treatments respectively increased and attenuated the expression of ERβ, supporting the proposed relationship between HIF-1α and ERβ. Finally, they found that KO of ERβ leads to a diminished response to E2 during hypoxia—giving further support to the protective properties of ERβ [49]. Nadadur et al. expanded on the protective properties of E2-mediated ERβ activation, stating that it serves to reverse adverse fibrosis and RV remodeling associated with monocrotaline (MCT)-induced PH in male and female rats [33]. They concluded that these therapeutic effects are likely a result of the ERβ-mediated upregulation of two novel extracellular matrix-degrading a disintegrin and metalloproteinases (ADAMs), ADAM15 and ADAM17, as well as osteopontin [33]. Umar et al. also explored E2-mediated PH protection in aging apolipoprotein E (ApoE)-deficient female mice in MCT model. They found that E2 therapy is able to rescue PH in these mice and restore the expression of ERβ to normal from its lowered state in ApoE-deficient mice. While the underlying mechanism remains unclear, this further suggests that ERβ can protect against, and even reverse, experimental PH [52].

Studies also suggest that ERα and ERβ cooperate in order to confer protection against PH [53,54]. For example, Lahm et al. demonstrated that both ERα-agonist PPT and ERβ-agonist DPN decrease pulmonary artery vasoconstriction in adult male Sprague-Dawley rats. They discovered that the effects of both ERs were attenuated in the presence of a nitric oxide synthase (NOS) inhibitor, which suggests that NO plays a vital role in mediating the pulmonary vascular effects of both ERα and ERβ [54]. In another study, Lahm et al. investigated the therapeutic effects of E2 receptors in male Sprague-Dawley rats with hypoxia-induced PH. Introduction of an ERα- but not ERβ-antagonist diminished the restorative effects of E2 on RV hypertrophy and cardiac output. However, both ERα- and ERβ-specific antagonists attenuated pulmonary artery remodeling, suggesting that both are involved in the process. This study also showed that E2 reduced activation of the ERK1/2 pathway and increased expression of cell-cycle inhibitor p27Kip1, which mediates cell proliferation in hypoxic PH [53]. 

GPR30 has also been shown to mediate the protective effects of E2 against PH [55,56]. Alencar et al. studied the impact of GPR30 in male Wistar rats with MCT-induced PH [55]. After treating the rats with GPR30-selective agonist G1, they noted reversal of PH-related cardiopulmonary abnormalities—including reduced pulmonary flow, RV hypertrophy, increased RV systolic pressure, and left ventricular (LV) dysfunction. These effects are achieved through promoting the synthesis of pulmonary endothelial NO, regulating Ca2+ handling, reducing cardiomyocyte inflammation, and decreasing collagen deposition in pulmonary and cardiac fibroblasts [55]. In a follow-up study, Alencar et al. analyzed the effects of G1 on OVX female rats with MCT-induced PH, and found similar results: the treatment reverses PA dysfunction, reduces RV overload, decreases RV dilation, reverses wall hypertrophy, reduces RV collagen deposition, and normalizes LV dysfunction [57]. 

In summary, ERα activation has been shown to attenuate vasoconstriction and RV hypertrophy. However, ERα has also been shown to increase PASMC proliferation and vascular remodeling, and its role in RV systolic pressure is controversial. ERβ decreases fibrosis, inflammation, vasoconstriction, RV remodeling, and RV hypertrophy. Finally, GPR30 decreases collagen deposition, vascular dysfunction, inflammation, RV overload, and RV dilation (Table 2, Figure 1). 

### 2.3. Atherosclerosis

Atherosclerosis is characterized by the narrowing of arterial walls due to plaque build-up [58]. Although more young men die from atherosclerosis and related diseases than young women, atherosclerosis-related diseases are the primary cause of death in postmenopausal women [59,60]. It is hypothesized that this phenomenon is due in part to the loss of E2 reactivity, suggesting that the atheroprotective properties of E2 receptors may be sex-specific in nature [59]. 

Studies have shown the atheroprotective role of ERh [61,62]. Foundationally, Billon-Galés et al. found that endothelial (and not hematopoietic) ERα plays a crucial role in the E2-mediated effects against atherosclerosis in low-density lipoprotein (LDL) receptor-deficient mice [61]. Here, hepatocyte deletion of ERα in female mice increased serum cholesterol levels and HDL particle sizes, ultimately resulting in increased atherosclerotic lesions. Furthermore, Zhu et al. found that hepatocyte ERα signaling is crucial to reverse cholesterol transport, and protects against arterial lipid accumulation during early atherosclerosis development in female mice fed a Western (high-cholesterol) diet [62]. This was determined by measuring the removal of ^3^H-cholesterol from macrophages to the liver, and subsequently to feces. Interestingly, Campesi et al. reported that the presence of lipopolysaccharides, which are known to increase the morbidity and mortality of atherosclerotic diseases, give rise to a sex-specific difference in human blood-monocytes derived macrophage ERα activation, with males showing a significantly higher level of expression than females [63]. 

Min et al. proposed a mechanism by which decreased ERα expression can interfere with E2 regulation of VSMC proliferation in ApoE/Leptin receptor double KO female mice. They suggest that the downregulation of ERα in VSMC occurs due to methylation of the second exon region of the ERα gene, which in turn is caused by increased expression of DNA methyltransferases in high-insulin conditions [64]. In a seemingly converse relationship, Ortmann et al. found that ERα itself inhibits the proliferation of VSMC in high-glucose conditions [65]. Considering that VSMC are a major source of reactive oxygen species (ROS) in the vasculature, and that ROS play a role in advanced atherosclerosis, this is yet another example of the atheroprotective role of ERα [66]. Billon-Galés et al. expanded on the mechanistic role of ERα, suggesting that while both AF1 and AF2 of ERα are necessary for its uterotrophic function, only AF2 is necessary for its aforementioned atheroprotective abilities [67]. However, there exists some controversy in that Smirnova et al. found that AF1-mediated nuclear action is necessary to restrict post-injury VSMC proliferation [29]. Yang et al. proposed an alternate mechanism that instead focuses on the antioxidant properties of the ERα-mediated PI3K/Akt pathway [68]. Here, they found that the introduction of GP-17, a phytoestrogen agonist of ERα, reduced blood lipid levels and upregulated the PI3K/Akt pathway, thereby decreasing the formation of atherosclerotic lesions in ApoE KO male mice. In summary, the atheroprotective properties of ERα are well-studied, and there exist a plethora of explanations for this phenomenon—ranging from its role in VSMC differentiation to the antioxidant nature of related pathways.

A study conducted by Villablanca et al. has suggested ERα mediates the susceptibility to atherosclerosis development in male mice [69]. They found that ERα-wildtype (WT) male mice fed a high-fat, high-cholesterol diet showed increased susceptibility to early atherosclerosis development and more extensive atherosclerotic lesion areas and distribution compared to ERα-KO mice [69]. Here, they suggested that testosterone mediates the ERα-dependent atheroma in male mice. 

Though the mechanisms by which ERα protects against the development of atherosclerosis are widely studied, the role of ERβ remain less thoroughly investigated and controversial [63,70]. Similar to ERα, ERβ is known to be expressed in both endothelial and VSMC of arteries [70,71]. Villard et al. noted decreased expression of ERβ in aneurysmal aorta compared to unaffected aorta in both men and women [72]. However, considering that atherosclerosis is only one cause of aneurysms and that Villard et al. also noted an increase in androgen receptors, the conclusive strength of this finding depends on further investigation. Interestingly, Christian et al. found that increased ERβ expression was positively correlated with calcification and atherosclerosis in the intimal layer of coronary arteries recovered from autopsies of both pre- and post-menopausal women, which suggests a potential controversy in the atheroprotective role of ERβ [70]. Further fueling this controversy is the fact that Campesi et al. also noted a decrease in ERβ expression in female human blood-monocytes derived macrophages exposed to lipopolysaccharides [63]. Using methylation specific polymerase chain reaction and combined bisulfite restriction analysis of the promoter region of the ERβ gene, Kim et al. offered an epigenetic perspective on the decreased expression of ERβ in atherosclerosis patients. Here, they found that coronary atherosclerotic tissues showed significantly higher methylation levels than healthy arterial and venous tissues [73]. Higher levels of methylation were observed in plaque regions of the ascending aorta, common carotid artery, and femoral artery of two patients compared to non-plaque regions. This suggests that epigenetic modification in the ERβ gene may give rise to vascular aging and atherosclerosis [73]. In summary, although there is evidence in support of the atheroprotective properties of ERβ, the aforementioned controversial findings suggest a need for further research into the seemingly sex-specific properties of ERβ atheroprotection.

Interestingly, in aged male and female ApoE KO mice, which are prone to develop advanced atherosclerosis, E2 has been shown to increase calcification in advanced atherosclerotic plaques by promoting the differentiation of VSMC to osteoblast-like cells [74]. Additionally, antagonism or silencing of ERα or ERβ was shown to promote the differentiation of VSMC to osteoblast-like cells, resulting in increased calcification. Further research is required in order to conclude whether ERa and ERβ protect against atherosclerosis [74]. 

GPR30 has been shown to mediate the protective effects of E2 against atherosclerosis [75,76]. In ovary-intact mice, Meyer et al. found that KO of GPR30 worsened the progression of atherosclerosis, increased LDL cholesterol levels and inflammation, and reduced vascular NO bioactivity [75]. They also confirmed the atheroprotective effects of GPR30 by treating mice with GPR30 agonist G1, which reduced postmenopausal atherosclerosis. Furthermore, Barton et al. reported that deletion of GPR30 increases endothelium-dependent vasoconstriction, visceral obesity, LDL levels, and inflammation [76]. Furthermore, Li et al. discovered that the activation of GPR30 inhibits the proliferation of coronary artery smooth muscle cells by suppressing the progression of the cell cycle [77]. Finally, Ghaffari et al. reported that E2 signaling through GPR30 via the transactivation of EGFR significantly inhibits the transcytosis of LDL by decreasing expression of endothelial scavenger receptor class B type 1 [78]. They also found that the expression of GPR30 was higher in endothelial cells than in hepatocytes [78]. In summary, though the atheroprotective function of GPR30 is a relatively novel field of study, the existing literature offers explanations for this role unique from those for ERα and ERβ. 

To conclude, various estrogen receptors protect against the onset and development of atherosclerosis. There is a wide array of evidence supporting the atheroprotective properties of ERα, such as a decrease in VSMC differentiation, increase in antioxidant properties, and a decrease in lipid accumulation. ERβ activation has been shown to decrease calcification and VSMC differentiation. Lastly, GPR30 activation decreases inflammation, suppresses CASMC proliferation, and attenuates vasoconstriction (Table 3, Figure 1). 

## 3. Role of Estrogen Receptors in Heart Pathology 

### 3.1. Ischemia/Reperfusion Injury

Myocardial ischemia-reperfusion (I/R) injury is the tissue damage caused by reperfusion of the myocardium following a period of ischemia [79]. I/R injury can occur as a result of inflammation, oxidative stress, intracellular and mitochondrial calcium overload, apoptosis, or necrosis [80,81]. During reperfusion, an abundance of cellular enzymes get released, which worsens the damage already sustained by the myocardium exposed to ischemia alone [80]. I/R injury contributes to adverse cardiovascular outcomes following myocardial ischemia, cardiac surgery, or circulatory arrest [79]. Animal studies have shown pre-menopausal females to have cardioprotection against I/R injury compared to males [82,83,84]. E2 is known to be a critical determinant of cardiovascular sex differences against I/R injury [10,85]. 

Studies have suggested the cardioprotective role of ERα in the regulation of I/R injury [82,86]. In an in vivo female rabbit model, activation of ERα by an ERα-specific agonist PPT reduced myocardial damage after regional I/R by reducing infarct size and inhibiting the complement system [86]. Here, the cardioprotective effects were not observed upon the administration of the ERβ-specific agonist DPN. Isolated female mouse hearts with an overexpression of ERα were protected against I/R injury compared to wild type female mice [82]. An increase in protective ERK1/2 activation and a decrease in pro-apoptotic c-Jun N-terminal kinase (JNK) activation was thought to play a role in the cardioprotective effects of ERα. It has also been shown in male mice that KO of ERα is associated with more severe damage following I/R injury [87]. Males with ERα KO had impaired mitochondrial respiratory function, decreased nitrite production, and significantly higher calcium accumulation compared with control hearts during reperfusion.

The cardioprotective role of ERβ has been highlighted in the recent years [88,89]. Isolated female mouse hearts lacking ERβ are less protected against I/R injury compared to wild-type female mice [88]. Activation of ERβ via a specific ERβ agonist prior to I/R has been shown to improve cardiac recovery by decreasing apoptosis and preserving mitochondrial integrity in female mice in vivo [90]. Here, anti-apoptotic protein levels of both B-cell lymphoma 2 and acetyl coenzyme A acyltransferase 2 were increased, and caspase 9 levels were decreased following I/R in hearts treated with ERβ agonist compared to controls. In another study, pre-treatment with ERβ-agonist DPN for two weeks resulted in cardioprotective effects against I/R injury in isolated hearts of OVX female mice, potentially through upregulating cardioprotective genes such as the anti-apoptotic protein, heat shock protein, cyclooxygenase 2, and growth arrest and DNA damage 45 beta [91]. 

It is important to consider the possibility that perhaps both ERα and ERβ are exerting cardioprotective effects against I/R injury to some extent. In cultured neonatal rat cardiomyocytes stimulated with I/R in the form of hypoxia/reoxygenation, it has been shown that E2 exerted cardioprotective effects by inhibiting ROS generation and preventing tumor protein p53 (p53)-dependent apoptosis [92]. Here, both ERα and ERβ contribute to p53 inhibition and cell survival. 

A study conducted on both male and female rats showed that acute activation of GPR30 with G1 is cardioprotective against I/R injury by reducing infarct size [9]. The G1-mediated cardioprotection is mediated by the activation of the PI3K/Akt pathway, and this cardioprotection dissipates in both genders with the administration of wortmannin, an inhibitor of the PI3K pathway [9]. Whether G1 treatment is also mediated by the activation of the ERK pathway is unclear. In the previous study, although a significant increase was shown in ERK phosphorylation following the G1 treatment, this cardioprotection was shown to occur independently of ERK activation, as blocking ERK via PD-98059 did not diminish the cardioprotective effects of G1 [9]. However, another study showed that the cardioprotective effects of G1 in I/R injury is dependent on the phosphorylation of the ERK pathway by inhibiting Ca^2+^-induced mitochondrial permeability transition pore opening, which plays a critical role in cell death after I/R injury [93]. Here, in the presence of PD-98059, the protective role of G1 on the heart function recovery, infarct size, and the mitochondrial calcium retention capacity were all eradicated. Lastly, it was shown in a recent study that GPR30 activation post-I/R activates the MEK/ERK pathway, which in turn deactivates the glycogen synthase kinase 3 (GSK-3β) pathway. This leads to the reduction of mitochondrial protein ubiquitination that leads to the inhibition of mitochondrial permeability transition pore opening and finally, a reduction in ROS production [94]. Thus, the precise role of pathways involved in the cardioprotective effects of GPR30 stimulation warrants further investigation. 

Taken together, all three ERs have been shown to contribute to myocardial protection following I/R. ERα has been shown to reduce infarct size, decrease ROS production, attenuate myocyte apoptosis, and inhibit the complement system. ERβ activation has been shown to attenuate apoptosis and decrease ROS production. Finally, GPR30 activation reduces infarct size and decreases ROS production (Table 4, Figure 2).

### 3.2. Heart Failure with Reduced Ejection Fraction

Heart failure with reduced ejection fraction (HFrEF), otherwise known as systolic heart failure (HF), is a disease characterized by a left ventricular EF of 40% or less [95]. Approximately 50% of all patients suffering from HF exhibit HFrEF [95]. The incidence of HFrEF is shown to be higher in men at any given age [96]. Considering these sex differences in the presentation and prognosis of HFrEF, E2 may play a protective role against its onset and development [97]. This section will serve to discuss the protective effects of ERs against HFrEF. 

Studies have delved into the physiological function and protective role of ERα against HFrEF [97,98]. Westphal et al. examined the ability of ERα to improve cardiac function in OVX female mice with myocardial hypertrophy induced by transverse aortic constriction (TAC) [98]. After administrating the mice with an ERα selective agonist 16α-LE2, E2, raloxifene, and placebo, they reported that the group receiving 16α-LE2 showed the smallest reduction in EF and improved systolic function nine weeks after TAC. Furthermore, histological analysis of cardiac tissue revealed that the groups treated with 16α-LE2 and E2 had a significantly lower fibrosis score (indicative of collagen content) compared to the raloxifene and placebo groups. Overall, this study demonstrated the ability of ERα to slow the progression of myocardial hypertrophy and reduce systolic dysfunction by preventing the reduction in ejection fraction (EF) [98]. Mahmoodzadeh et al. reported that the expression and localization of ERα are altered in both female and male humans with HF. In healthy hearts, ERα colocalizes with β-catenin at the intercalated discs to confer structural stability [99]. In patients with end-stage human dilated cardiomyopathy, a 1.8-fold increase in ERα mRNA and protein has been reported compared to controls. As dilated cardiomyopathy progresses, however, there is a loss of colocalization between ERα and β-catenin in intercalated discs. Here, they concluded that the increase in total ERα expression may represent a compensatory mechanism to contribute to the stability of cardiac intercalated discs, but that loss of its association with β-catenin is a significant contributor to the progression of HF [99]. 

Many studies have suggested the protective role of ERβ against HFrEF [100,101,102]. Kararigas et al. used the aforementioned TAC model to instead study the cardioprotective abilities of ERβ [100] They compared WT and ERβ-KO female mice and demonstrated that the deletion of ERβ leads to a significant increase in the activation of inflammatory pathways. Here, ERβ acts as a “gatekeeper” of the heart’s genomic response to hypertrophy induced by pressure overload; this indicates that ERβ could be a valid target for HF rescue [100]. Another study by Kararigas et al. suggested that the action of ERβ may be sex-specific in nature [101]. After comparing age-matched male and female WT and ERβ-KO mice, they found that ERβ displays a more prominent cardioprotective role in female mice. Furthermore, they showed that ERβ is crucial for modulation of the heart’s proteomic response to pressure overload, which can play an important role in preventing the onset of HF [101]. Here, they concluded that the observed differences can potentially identify sex-specific targets for the treatment of HF. Another study by Skavdahl et al. confirmed the seemingly sex-specific nature by which ERβ attenuates the hypertrophic response to pressure overload [102]. Using ERα- and ERβ-KO female mice that underwent TAC, they reported that ERα-KO mice show identical heart weight to body weight ratios compared to their WT counterparts two weeks post-surgery. However, ERβ-KO mice showed a significantly greater increase in heart weight to body weight ratio compared to WT littermates in response to TAC [102]. Together, these findings indicate the non-necessity of ERα and the importance of ERβ in the myocardial response to pressure overload. Notably, because WT males showed significantly higher hypertrophy levels in response to TAC than WT females, it is possible that the difference in ERβ expression could be partly or solely responsible for this disparity [102]. Fliegner et al. highlighted sex differences in the cardioprotection offered by ERβ in mice with TAC-induced pressure overload. Here, they found that both female sex and ERβ attenuate the development of fibrosis and apoptosis in response to pressure overload, and consequently, delay the progression to HF [103]. Furthermore, they reported that ERβ-KO males who received a sham operation had a significantly lower EF (40%) than similar females (55%), suggesting that ERβ contributes to the maintenance of EF in male hearts. Finally, they found that ERβ KO causes a sex-specific promotion of cardiomyocyte apoptosis—with male mice showing a significantly greater increase in pro-apoptotic gene expression than female mice. Together, these results support not only the cardioprotective properties of ERβ in both males and females, but also suggest that the differences in ERβ on cardiac remodeling between males and females can potentially influence the progression to HF in later stages in a sex-specific manner [103]. 

More recently, Iorga et al. highlighted the cardioprotective role of ERβ in male mice with TAC-induced HF [88]. After treating the mice with ERα-agonist PPT and ERβ-agonist DPN, they reported that only DPN is able to significantly improve EF. Furthermore, they found that E2 is unable to rescue HF in the presence of PHTPP, an ERβ-antagonist. Here, ERβ activation was shown to be associated with restoration of cardiac angiogenesis, suppression of cardiac fibrosis, and normalization of hemodynamic parameters. Together, these findings suggest that ERβ activation is responsible for rescue of pre-existing severe HF [88]. Considering that E2 replacement therapy upregulates the expression of ERβ more than ERα in OVX female Sprague-Dawley rat hearts [104], this selective responsiveness can be used as a tool for the development of therapeutic interventions against HF and other cardiac dysfunctions.

Studies have also delved into the protective role of GPR30 against pre-existing severe HF [104,105]. Kang et al. explored the ability of GPR30-agonist G1 to reduce isoproterenol-induced HF in female OVX Sprague-Dawley rats [106]. They found that treatment with G1 improves cardiac function, increases myocyte contractility, and reduces fibrosis. Moreover, they reported that the protective effects of GPR30 may be achieved through mediating the expression of β1- and β2-adrenergic receptors of ventricular myocytes. Here, treatment with β1- and β2-adrenergic receptors antagonists diminished the improvement of cardiac function achieved by G1 treatment, which led to the conclusion that G1 attenuates HF through normalizing β1- adrenergic receptor expression and increasing β2- adrenergic receptor expression [106]. Another study conducted by Delbeck et al. discovered that the protective effects of GPR30 against HF are sex-specific in nature [105]. They demonstrated that KO of GPR30 in male, but not female mice causes impaired LV function as well as LV dilatation. Hemodynamic measurements indicated decreased contractility and relaxation capacity of the LV, giving way to increased LV end-diastolic pressure in male KO mice [105]. 

In summary, various estrogen receptors protect against the onset and development of HFrEF. ERα activation reduces fibrosis, maintains EF, decreases hypertrophy, and increases systolic function. ERβ activation attenuates myocyte fibrosis, decreases apoptosis, stimulates cardiac angiogenesis, decreases inflammation, and normalizes hemodynamic parameters. Finally, GPR30 reduces apoptosis and ROS production (Table 5, Figure 2). 

### 3.3. Heart Failure with Preserved Ejection Fraction

Heart failure with preserved ejection fraction (HFpEF) is a disease characterized by symptoms of HF in the presence of a normal systolic function, as assessed by EF [107]. In turn, diastolic function such as myocardial relaxation is impaired [108]. The novelty and pathophysiological heterogeneity of this disease’s characterization has led to significant challenges in establishing universal terminology and animal models, generating an urgent need to better understand its pathophysiological mechanisms [106,107,108]. In the United States, HFpEF accounts for 50% of patients with HF, a majority of which are post-menopausal women [109,110,111]. As a result, many reviews have focused on the role of E2 in the development of HFpEF. However not much is known regarding the specific roles of the ERs, making it the focal point of current research [112]. Due to the aforementioned lack of established HFpEF animal models, this section will summarize the existing body of research on the relationship between the ERs and diastolic dysfunction, an important precursor to HFpEF. 

Though the role of ERα in HFpEF has not been widely studied, Cheng et al. delved into the protective role of ERα against diastolic dysfunction [113]. Male and female WT and ERα-mutant rats underwent pulmonary artery banding to induce RV hypertrophy. Here, they found that in females, ERα protects against diastolic dysfunction and collagen accumulation in the adaptation of the RV to pressure overload. Female, but not male ERα-mutant rats subjected to RV hypertrophy displayed diastolic dysfunction as well as an increased ratio of collagen type I to collagen type III fibers. This study suggests that the protective action of ERα against diastolic dysfunction may be sex-specific in nature [113]. 

ERβ has been shown to modulate the E2-mediated diastolic functional protective pathways [50,114]. Pedram et al. found that E2 inhibits AngII-induced cardiac hypertrophy, a precursor of diastolic stiffness, in female mice through ERβ as demonstrated by an ERβ-specific agonist 8β-VE2 [115]. This effect was not shown to be mediated by ERα, since activation of only ERβ was shown to attenuate cardiac hypertrophy. E2 binding to ERβ inhibited the AngII-induced transition from α to β myosin heavy chain production, ERK activation, calcineurin activity, and interstitial fibrosis. Here, left ventricular ejection fraction (LVEF) was shown to be unimpaired following AngII infusion. In addition to the aforementioned modulatory effects of E2 via ERβ, this complex has also been shown to upregulate NOS in neonatal rat cardiac myocytes [114]. Jessup et al. found that neuronal NOS inhibition improved diastolic function in OVX rats [116]. E2 is thought to regulate the production of NOS via activation of tetrahydrobiopterin, a cofactor necessary for NOS activity. For this reason, a lack of sufficient E2 may have led to neuronal NOS uncoupling in OVX rats, resulting in a transition from production of NO to superoxide [116]. Future studies should identify whether ERβ-specific inhibition can prevent this uncoupling and restore diastolic function. 

The majority of work regarding the role ERs play in attenuating diastolic dysfunction is focused on GPR30 activation of different pathways [114,115,116,117]. Da Silva et al. found that administration of G1, a GPR30 selective agonist played a paramount role in improving diastolic function in OVX spontaneously hypertensive rats. Chronic G1 was also shown to improve the vasorelaxant responsiveness to acetylcholine aortic rings in vitro [117]. This increased lusitropy conferred by G1 was linked to decreases in cardiac angiotensin-converting enzyme, angiotensin type I receptor expression, and AngII immunoreactivity, suggesting GPR30 activation modulates the renin-angiotensin system (RAS) [117]. Alencar et al. proposed an alternative mechanism in which GPR30 ameliorates impaired myocardial relaxation through increased cardiomyocyte sarcoplasmic reticulum Ca^2+^ ATPase expression and decreased cardiac fibrosis in aging OVX female rats [118]. Furthermore, Wang et al. found that chronic GPR30 activation via G1 administration prevented increases in cardiac NAD(P)H oxidase (NOX4) expression in OVX rats, potentially leading to conserved diastolic function [119]. NOX4 expression is increased in response to hypertrophic stimuli and is known to be a major source of oxidative stress through ROS production [119]. Therefore, GPR30′s preservation of diastolic function may be linked to modulation of NOX4 expression, prompting future studies in order to elucidate this relationship. Jessup et al. also used rats with an aim to investigate the effect of G1 on salt-induced diastolic dysfunction [120]. Tissue Doppler studies showed that G1 administration improved myocardial relaxation in both normal salt and high salt groups as shown by an increase in mitral annular descent. Moreover, high salt rats treated with G1 had an increased ratio of early to late LV filling. Interestingly, GPR30 mRNA expression was increased by 80% in the high salt rats versus the normal salt rats, suggesting further investigations are required in order to elucidate the role of GPR30 in salt-induced diastolic dysfunction [120]. 

Although G1 has shown to improve diastolic function, it is difficult to draw cardiac-specific conclusions regarding G1 administration due to the fact that GPR30 activation modulates numerous pathways throughout the body. Noting this knowledge gap, Wang et al. generated a cardiomyocyte specific GPR30 KO mouse model to investigate cardiac specific effects of GPR30 gene deletion [8]. Here, they found that GPR30 KO mice exhibited diastolic dysfunction, along with other characteristics of CVD, thus not only confirming the protection of GPR30 in lusitropy but also in cardiac function in general.

It is also important to note the possibility that E2 does not affect the protective outcomes observed in female mice. Recently, Tong et al. highlighted the protective effects of the female sex in a novel preclinical HFpEF model [121]. Here, male and female mice were induced to HFpEF via a combination of high-fat diet coupled with inhibition of constitutive NOS with N*ω*-Nitro-l-arginine methyl ester. The results of this study indicated that the female protection is likely not mediated by sex hormones, as ovariectomy compared to sham surgery on 8-week old female mice followed by HFpEF treatment for 15 weeks did not yield differences in diastolic function or heart weight [121]. Thus, future studies should recognize the possibility of crosstalk between E2 and other signaling factors in the cardioprotection of females against HFpEF. 

In summary, because HFpEF is a relatively novel disease state, characterization and representative animal models are still lacking. However, the limited studies on ERα and ERβ suggested that ERα reduces collagen deposition and dialostic dysfuction, and that ERβ reduces fibrosis and cardiac hypertrophy. Finally, extensive studies on GPR30 suggested that it is responsible for decreasing fibrosis and improving myocardial relaxation and diastolic function (Table 6, Figure 2). 

## 4. Future Perspectives and Concluding Remarks

In this review, we summarized current knowledge on the involvement of ERα, ERβ, and GPR30 in E2-mediated cardiovascular protection and their role in developing novel and effective therapeutic strategies for CVDs. 

Considering that premenopausal women have a lower rate of CVD than age-matched men, it is likely that sex hormones play a vital role in this disparity. Throughout the past three decades, the protective role of E2 in pre-clinical models of cardiovascular disease has been highlighted. E2 can signal through at least three separate receptors, which can in turn exert both genomic and non-genomic signaling to maintain cardiovascular homeostasis. The ERs can differentially modulate gene expression and thus exert the E2-mediated long-term effects through changes in cardiomyocyte gene expression [3]. The identity and effects of these target genes remain elusive. Furthermore, several studies have highlighted sex differences in the cardioprotection conferred by ERs, which can potentially influence the progression of CVD in later stages in a sex-specific manner [84,102]. 

Although the vast majority of studies have focused on the effects of gonadal hormones in sex differences in CVD, it is also important to take into account the potential effect of sex chromosome (XX vs. XY). In fact, throughout the past two decades, studies have shown that the type of sex chromosome (X vs. Y) as well as the number of X chromosomes (1 vs. 2) could also contribute to sex-dependent cardiovascular differences, even in the absence of gonadal hormones [122,123]. Genes expressed on the sex chromosomes have significant implications that contribute to sex differences in disease phenotypes, such as CVD, autoimmune disease, metabolic dysfunction, and neurodegeneration [122,123]. Whether sex chromosome genes can interact with gonadal hormones, such as E2, and in turn interact with its receptors remains to be seen. Though these studies are in their infancy, future research should tackle how the various sex-biasing factors (gonadal hormones, sex chromosome genes) might interact with one another in order to confer protection. 

To conclude, focusing on the mechanisms underlying sex-specific cardioprotection by the ERs will aid in the development of effective therapeutic strategies for future treatment of CVDs. There is a wide array of evidence supporting the cardioprotective properties of ERα, ERβ, and GPR30 in mediating the effects of E2. However, the exact contribution of each receptor in cardioprotection remains elusive. To date, studies with KO mice have shown conflicting results, which can potentially be due to the complex regulation of gene expression by these receptors. Despite the discrepancy between the possible cardioprotective functions of the ERs, it is necessary to recognize the possibility of crosstalk between the three ERs. In light of prior literature coming to competing conclusions regarding the individual efficacy of each receptor, it is important to consider the possibility that different ERs can interact with, or even compensate for, one another. It is, therefore, important that future work addresses not only this knowledge gap, but also how the various ERs might interact with one another in order to confer protection. 

## Figures and Tables

**Figure 1 ijms-21-04314-f001:**
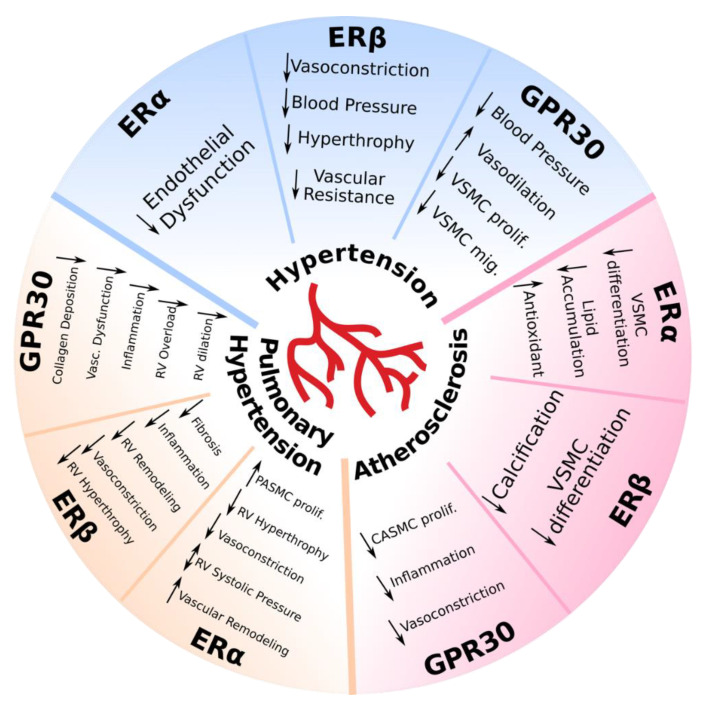
The role of estrogen receptors in vascular disease: hypertension, pulmonary hypertension, and atherosclerosis. Depicted pathways have been shown to be involved in animal models of cardiovascular disease. CASMC: coronary artery smooth muscle cells; ERα: estrogen receptor alpha; ERβ: estrogen receptor beta; GPR30: G-protein-coupled estrogen receptor; Mig: migration; PASMC: pulmonary artery smooth muscle cells; Prolif: proliferation; RV: right ventricular; Vasc: vascular; VSMC: vascular smooth muscle cells; ↑: increased; ↓: decreased; ↕: conflicting results.

**Figure 2 ijms-21-04314-f002:**
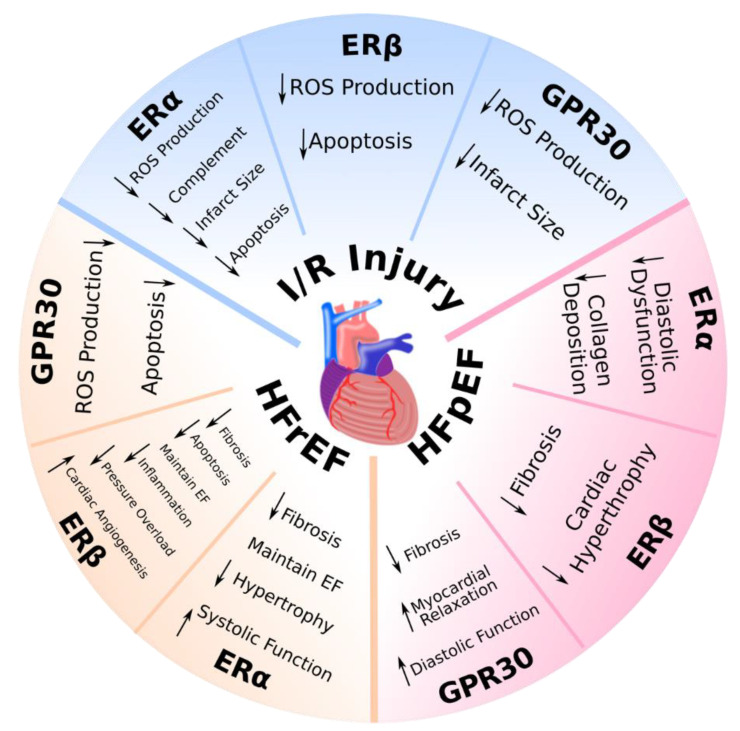
The role of estrogen receptors in heart disease: ischemia/reperfusion injury, heart failure with reduced ejection fraction, and heart failure with preserved ejection fraction. Depicted pathways have been shown to be involved in animal models of cardiovascular disease. EF: ejection fraction; ERα: estrogen receptor alpha; ERβ: estrogen receptor beta; GPR30: G-protein-coupled estrogen receptor; HFpEF: heart failure with preserved ejection fraction; HFrEF: heart failure with reduced ejection fraction; I/R injury: ischemia/reperfusion injury; ROS: reactive oxygen species; ↑: increased; ↓: decreased.

**Table 1 ijms-21-04314-t001:** Role of estrogen receptors in hypertension.

ER	Intervention	Model	Mechanism and Outcome	Ref
**ERα**	ERα KO	Ang II, OVX female mice	↑ sympathetic outflow → ↑ HTN	[21]
	ERα KO	Ang II, Female mice lacking membrane ERα	↓ Activation Factor 2-dependent transcription → ↑ HTN3	[22]
	ERα agonist	Spontaneously hypertensive OVX rats with reduced aortic eNOS	Normalized aortic eNOS → ↓ endothelial dysfunction	[23]
	ERα transfection	E2, Primary human VSMC	↓ iNOS activity → ↑ HTN	[24]
**ERβ**	ERβ KO	Male mice with VSMC ERβ KO	↑ abnormal vascular contraction, ↑ ion channel dysfunction, ↑ vasoconstriction, ↑BP → ↑ HTN	[24]
	ERβ agonist	Spontaneously hypertensive OVX rats	↓ BP, ↓ cardiac hypertrophy, ↓ vascular resistance → ↓ HTN	[25]
**GPR30**	GPR30 agonist	Human endothelial cells	↑ eNOS → ↑ c-Src/EGFR/PI3K/ERK → ↑ vasodilation → ↓ HTN	[30]
	GPR30 antagonist	Human endothelial cells	↓ NO production → ↓ vasodilation → ↑ HTN	[30]

Ang II: angiotensin II; BP: blood pressure; cSrc: proto-oncogene tyrosine-protein kinase; E2: 17beta-estradiol; EGFR: epidermal growth factor receptor; eNOS: endothelial nitric oxide synthase; ER: estrogen receptor; ERK: extracellular-signal-regulated kinase; Erα: estrogen receptor alpha; ERβ: estrogen receptor beta; GPR30: G protein–coupled receptor; HTN: hypertension; iNOS: inducible nitric oxide synthase; KO: knockout; NO: nitric oxide; OVX: ovariectomized; PI3K: phosphoinositide-3-kinase; VSMC: vascular smooth muscle cells; ↑: increased; ↓: decreased; →: results in.

**Table 2 ijms-21-04314-t002:** Role of estrogen receptors in pulmonary hypertension.

ER	Intervention	Model	Mechanism and Outcome	Ref
**ERα**	E2 repletion	OVX SuHx-induced female PH rats with decreased RV ERα expression	E2 repletion → ↑ ERα → ↓ RV systolic pressure → ↓ RV hypertrophy → ↓ PH	[42]
	ERα antagonist	Serotonin transporter with PH and mice with hypoxia-induced PH	↓ ERα → ↓ pulmonary vascular remodeling, ↓ RV systolic pressure, ↓ PASMC proliferation	[43]
	ERα agonist	Human PASMC in vitro	↑ ERα → ↑ Akt, ↑ MAPK → ↑ proliferation	[43]
	ERα transfection	Male mice lungs and OVX female mice lungs	↑ ERα in OVX females → ↓ BMPR2 gene expression → ↑ PH	[44]
**ERβ**	ERβ agonist	Male rats with monocrotaline-induced PH	↑ ERβ → ↓ fibrosis, ↓ inflammation, ↓ RV hypertrophy → ↓ PH	[39]
	Deferoxamine, HIF-1α stabilizer	Male rat lungs with hypoxia-induced PH	↑ hypoxia → ↑ ERβ in lung → ↓ HIF-1α, ↓ pulmonary vascular remodeling → ↓ PH	[49]
	ERβ KO	Male rat lungs with hypoxia-induced PH	↑ HIF-2α → ↓ response to E2 → ↑ PH	[49]
	E2 therapy	Male and female rats with MCT-induced PH	↑ E2 → ↑ ERβ → ↓ fibrosis, ↑ ADAM15/ADAM17/osteopontin the RV → ↓ RV remodeling	[33]
	E2 therapy	Female ApoE deficient mice with MCT-induced PH	↑ E2 → restore ERβ → ↓ PAH	[52]
**ERα and ERβ**	ERα agonist and EEβ agonist	Adult male rats	↑ ERα and ERβ → ↓ pulmonary artery vasoconstriction (attenuated with NOS inhibitor)	[54]
	ERα antagonist and ERβ antagonist	Male rats with hypoxia-induced PH	↓ ERα and ERβ → ↓ pulmonary RV remodeling	[53]
	E2 treatment	Male rats with hypoxia-induced PH	↑ E2 treatment → ↓ ERK1/2 activation in the lung and RV (attenuated with co-treatment of ERα and ERβ antagonist)	[53]
**GPR30**	GPR30 agonist	Male rats with MCT-induced PH	↑ GPR30 → ↑ eNOS, ↓ collagen deposition in pulmonary and cardiac fibroblasts, ↑ Ca2+ handling regulation and ↓ inflammation in cardiomyocytes → ↑ pulmonary flow, → ↑ RV hypertrophy, ↑ LV dysfunction	[55]
	GPR30 agonist	OVX female rats with MCT-induced PH	↑ GPR30 → ↓ pulmonary artery dysfunction, ↓ RV overload, ↓ RV dilation, ↓ wall hypertrophy, ↓ collagen deposition, normalizes LV dysfunction	[57]

ADAM15: a disintegrin and metalloproteinase 15; ADAM17: a disintegrin and metalloproteinase 17; Akt: protein kinase B; ApoE: apolipoprotein E-deficient; BMPR2: bone morphogenetic protein receptor type 2; Ca^2+:^ calcium; E2: 17beta-estradiol; eNOS: endothelial nitric oxide synthase; ER: estrogen receptor; ERK: extracellular signal-regulated protein kinases; ERK1/2: extracellular signal-regulated protein kinases 1 and 2; Erα: estrogen receptor alpha; ERβ: estrogen receptor beta; GPR30: G protein–coupled receptor; HIF-2α: hypoxia-inducible factor 2α; HIF-1α: hypoxia-inducible factor α; KO: knockout; LV: Left ventricle; PASMC: pulmonary artery smooth muscle cell; MAPK: mitogen-activated protein kinase; MCT: monocrotaline; NOS: nitric oxide synthase; OVX: ovariectomized; PAH: pulmonary arterial hypertension; PH: pulmonary hypertension; PI3K: phosphoinositide-3-kinase; RV: right ventricular; Su/Hx: SU5416/hypoxia-induced pulmonary hypertension; ↑: increased; ↓: decreased; →: results in.

**Table 3 ijms-21-04314-t003:** Role of estrogen receptors in atherosclerosis.

ER	Intervention	Model	Mechanism and Outcome	Ref
ERα	ERα KO	Male mice fed a high-fat, high-cholesterol diet	↑ ERα → ↑ atheroma formation → ↑ susceptibility to early atherosclerosis & more extensive atherosclerotic lesions	[69]
	Hepatocyte Erα deletion	Female mice fed a Western-type diet	↓ ERα → ↑ serum cholesterol levels, increased HDL particle sizes → ↑ size of early atherosclerotic lesions	[62]
	NA	Human blood monocytes-derived macrophages	↑ lipopolysaccharide-mediated inflammatory responses → ERα activation in males	[63]
	E2 treatment	OVX mice	↑ ERα → ↑ Activation Factor 2 → ↑ atheroprotection	[67]
	ERα agonist	ApoE KO male mice	↑ ERα → ↓ serum lipid levels, ↑ PI3K/Akt pathway → ↑ atherosclerotic lesions	[68]
ERβ ERα and ERβ	NA	Coronary arteries of pre- and post-menopausal women	↑ ERβ → ↑ coronary calcification → ↑ atherosclerosis	[70]
	NA	Coronary atherosclerotic tissues	Epigenetic modification in the ERβ gene → ↑ methylation levels → ↓ vascular aging, → ↑ atherosclerosis	[73]
	E2 treatment	ApoE KO male and female mice	↓ ERα and ERβ → ↑ differentiation of VSMC to osteoblast-like cells → ↑ calcification of advanced atherosclerotic lesions	[74]
GPR30	GPR30 KO	Ovary-intact mice with GPER deletion	↓ GPR30 → ↑ LDL cholesterol levels, ↑ inflammation, ↓ vascular NO bioactivity → ↑ progression of atherosclerosis	[75]
	GPR30 deletion	Male and female GPR30 KO mice	↓ GPR30 → ↑ endothelium-dependent vasoconstriction, ↑ visceral obesity, ↑ LDL levels, ↑ inflammation	[76]
	NA	Human coronary artery endothelial cells	↑ GPR30 → ↑ EGFR → ↓ endothelial scavenger receptor class B type I → ↓ LDL transcytosis	[78]

Akt: protein kinase B; ApoE: apolipoprotein E-deficient; E2: 17beta-estradiol; EGFR: epidermal growth factor receptor; ER: estrogen receptor; Erα: estrogen receptor alpha; ERβ: estrogen receptor beta; GPR30: G protein–coupled receptor; HDL: high-density lipoproteins; KO: knockout; LDL: low-density lipoproteins; NO: nitric oxide; PI3K: phosphoinositide-3-kinase; VSMC: vascular smooth muscle cell; ↑: increased; ↓: decreased; →: results in.

**Table 4 ijms-21-04314-t004:** Role of estrogen receptors in ischemia/reperfusion injury.

ER	Intervention	Model	Mechanism and Outcome	Ref
ERα	ERα agonist	Female rabbit model after regional I/R	↑ ERα → ↓ infarct size → ↓ complement system	[86]
	ERα overexpression	Isolated female mouse hearts after I/R	↑ ERα → ↑ ERK1/2 activation → ↓ pro-apoptotic JNK → ↑ cardioprotection	[82]
	ERα KO	Male mice after I/R	↓ ERα → impaired mitochondrial respiratory function, ↓ nitrite production, ↑ Ca2+ accumulation	[87]
ERβ	ERβ agonist	Female mice prior to I/R	↑ ERβ → ↓ apoptosis, preserved mitochondrial integrity, ↑ B-cell lymphoma 2, ↑ acetyl coenzyme A acetyltransferase 2, ↓ caspase 9 → ↑ cardiac recovery	[90]
	ERβ agonist pretreatmen	Isolated OVX female mouse hearts	↑ ERβ → ↑ cardioprotective genes (anti-apoptotic protein, heat shock protein, cyclooxygenase 2, growth arrest and DNA damage 45 beta)	[91]
ERα and ERβ	E2 treatment	Cultured neonatal rat cardiomyocytes stimulated with hypoxia/reoxygen	↑ ERα and ERβ → ↓ ROS, ↓ p53 dependent apoptosis → ↑ cardioprotection	[92]
GPR30	Acute activation with GPR30 agonist	Male and female rats during I/R injury	↑ GPR30 → ↑ PI3K/Akt pathway → ↑ cardioprotection	[9]
	GPR30 agonist	Isolated hearts from male mice undergoing I/R via Langendorff	↑ GPR30 → ↑ ERK pathway → ↓ Ca2+-induced mitochondrial permeability pore opening → ↓ cell death	[93]
	GPR30 agonist	In vivo rat hearts subjected to I/R	↑ GPR30 → ↑ MEK/ERK → ↓ GSK-3β pathway → ↓ mitochondrial dysfunction and mitophagy	[94]

Akt: protein kinase B; Ca^2+^: calcium; ERK: extracellular signal-regulated protein kinase; ERK1/2: extracellular signal-regulated protein kinases 1 and 2; ER: estrogen receptor; Erα: estrogen receptor alpha; ERβ: estrogen receptor beta; GPR30: G protein–coupled receptor; GSK-3β: glycogen synthase kinase 3; I/R: ischemia/reperfusion; JNK: c-Jun N-terminal kinase; KO: knockout; MEK: mitogen-activated protein kinase; NO: nitric oxide; OVX: ovariectomized; p53: tumor protein p53; PI3K: phosphoinositide-3-kinase; ROS: reactive oxygen species; ↑: increased; ↓: decreased; →: results in.

**Table 5 ijms-21-04314-t005:** Role of estrogen receptors in heart failure with reduced ejection fraction.

ER	Intervention	Model	Mechanism and Outcome	Ref
ERα	ERα agonist	OVX female mice with myocardial hypertrophy induced by TAC	↑ ERα → smallest reduction in EF, ↓ fibrosis → improved systolic function	[98]
	NA	Male and female humans with end-stage HF	Human HF → loss of colocalization between Erα and β-catenin, altered expression and localization of ERα	[99]
ERβ	ERβ KO	Female mice induced by TAC	↓ ERβ → ↑ inflammatory pathways	[100]
	ERβ KO	Male and female WT vs. ERβ KO mice	↑ ERβ = more prominent cardioprotective role in females → ↓ pressure overload	[101]
	ERβ KO	ERβ KO mice that underwent TAC	↓ ERβ → ↑ hypertrophy	[102]
	ERβ KO	Female mice with TAC-induced pressure overload	↓ ERβ → ↑ cardiac fibrosis, ↑ apoptosis → ↑ HF	[103]
	ERβ agonist	Male mice with TAC-induced HF	↑ ERβ → ↓ cardiac fibrosis, ↑ EF, restoration of cardiac angiogenesis, normalization of hemodynamic parameters	[88]
GPR30	GPR30 agonist	Female OVX rats with isoproterenol-induced HF	↑ GPR30 → mediates the expression of β1- and β2-adrenergic receptors → ↑ cardiac function, ↑ myocyte contractility, ↓ fibrosis	[106]
	GPR30 KO	Male GPR30-deficient mice	↓ GPR30 → impaired cardiac function in LV, LV enlargement, ↓ contractility/relaxation of LV → ↑ LV end-diastolic pressure	[105]

EF: ejection fraction; ER: estrogen receptor; Erα: estrogen receptor alpha; ERβ: estrogen receptor beta; GPR30: G protein–coupled receptor; HF: heart failure; KO: knockout; LV: left ventricular; OVX: ovariectomized; TAC: transverse aortic constriction; WT: wild type; ↑: increased; ↓: decreased; →: results in.

**Table 6 ijms-21-04314-t006:** Role of estrogen receptors in heart failure with preserved ejection fraction.

ER	Intervention	Model	Mechanism and Outcome	Ref
ERα	ERα mutant	Male and female rats via pulmonary artery banding-induced RV hypertrophy	↓ ERα in females → ↑ collagen type I → ↑ diastolic dysfunction	[113]
ERβ	ERβ agonist	Ang II-induced cardiac hypertrophy in female mice	↑ ERβ → ↓ transition from α to β myosin heavy chain production, ↓ ERK activation, ↓ calcineurin activity, ↓ interstitial fibrosis	[115]
	ERβ antagonist	Neonatal rat cardiac myocytes transfected with ERβ antagonist	↑ ERβ → ↑ expression of eNOS and iNOS	[114]
GPR30	GPR30 agonist	OVX spontaneously hypertensive rats	↑ GPR30 → ↓ cardiac angiotensin-converting enzyme, ↓ angiotensin type I receptor expression, ↓ AngII immunoreactivity → improved vasorelaxant responsiveness	[117]
	GPR30 agonist	Aged OVX female rats	↑ GPR30 → ↑ cardiomyocyte sarcoplasmic reticulum Ca2+ ATPase expression, ↓ cardiac fibrosis → ameliorates impaired myocardial relaxation	[118]
	Chronic GPR30 agonist	OVX female rats	↑ GPR30 → prevents increases in cardiac NOX4 expression → conserved diastolic function	[119]
	GPR30 agonist	Salt-induced diastolic dysfunction in female rats	↑ GPR30 → ↑ myocardial relaxation, ↑ ratio of early-to-late LV filling	[120]
	GPR30 KO	Cardiomyocyte specific GPR30 KO mice	↓ GPR30 → ↑ LV dysfunction, adverse cardiac remodeing	[8]

Ang II: angiotensin II; eNOS: endothelial nitric oxide synthase; ER: estrogen receptor; ERK: extracellular signal-regulated protein kinase; Erα: estrogen receptor alpha; ERβ: estrogen receptor beta; GPR30: G protein–coupled receptor; iNOS: inducible nitric oxide synthase; KO: knockout; LV: left ventricular; NOX4: NAD(P)H oxidase; OVX: ovariectomized; RV: right ventricular; ↑: increased; ↓ decreased; →: results in.

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
