# Peer review of "The Role of Estrogen Receptors in Cardiovascular Disease"

_ijms, 2020, doi:10.3390/ijms21124314_

Round 1

Reviewer 1 Report

In their manuscript entitled, ‘The Role of Estrogen Receptors in Cardiovascular Disease’, Aryan et al. are reviewing the current understanding of the underlying molecular mechanisms of estrogen receptors in regulating vascular pathology, with a special emphasis on hypertension, pulmonary hypertension, and atherosclerosis, as well as in regulating cardiac pathology. The authors are putting a particular emphasis on ischemia/reperfusion injury, heart failure with reduced ejection fraction, and heart failure with preserved ejection fraction and are summarizing current knowledge on the involvement of ERα, ERβ, and GPR30 in E2-mediated cardiovascular protection and their role in developing novel and effective therapeutic strategies for CVDs.

Overall, the manuscript addresses an important question regarding underlying molecular mechanisms in sex differences regarding cardiovascular disease, focusing on the effect of estrogen receptors in CVD.

The review article is highly accurate. While it is challenging to follow at times, the communicated content is educative and well-integrated.

Specific considerations:

  • Seeing the length of the manuscript, I could imagine that more sub-headings would improve the structure and thus improve navigation for the reader.
  • In Figure 1 the direction of writing requires the reader to turn the screen, which negatively impacts the function of the figure – providing an easy-access overview of the content. I strongly suggest to provide the writing in a horizontal fashion. The same accounts for Figure 2.
  • In lines 79/80 I suggest to revise the definition of hypertension according to current clinical guidelines (i.e. AHA or ESC guidelines) and cite accordingly.
  • The authors state that hypertension is higher amongst women in individuals aged >60 years. The cited publication ‘16’ reports results specific to Korea. Is there any further data in this respect? Otherwise I suggest that the statement needs to be tuned down, since it seems a bit surprising.
  • In line 177, the authors first mention BMPR2. I suggest to provide an introductory sentence to this receptor, which explains the connection to the topic it is used under.
  • Please check first mention of PASMC and its initial explanation.
  • Please check authors in reference ‘13’

Author Response

Response to Reviewer #1:

In their manuscript entitled, ‘The Role of Estrogen Receptors in Cardiovascular Disease’, Aryan et al. are reviewing the current understanding of the underlying molecular mechanisms of estrogen receptors in regulating vascular pathology, with a special emphasis on hypertension, pulmonary hypertension, and atherosclerosis, as well as in regulating cardiac pathology. The authors are putting a particular emphasis on ischemia/reperfusion injury, heart failure with reduced ejection fraction, and heart failure with preserved ejection fraction and are summarizing current knowledge on the involvement of ERα, ERβ, and GPR30 in E2-mediated cardiovascular protection and their role in developing novel and effective therapeutic strategies for CVDs.

Overall, the manuscript addresses an important question regarding underlying molecular mechanisms in sex differences regarding cardiovascular disease, focusing on the effect of estrogen receptors in CVD.

The review article is highly accurate. While it is challenging to follow at times, the communicated content is educative and well-integrated.

We thank the Reviewer for acknowledging the importance and great interest of our work to the scientific community. We also appreciate the time and effort that have been taken to provide constructive comments on our work.

  1. Seeing the length of the manuscript, I could imagine that more sub-headings would improve the structure and thus improve navigation for the reader. We thank the reviewer for their comment. We would like to state that we did, in fact, try adding additional subheadings at first. However, after doing so, we realized that it would inundate the reader. That is why we have chosen to adhere to the original sub-headings. However, in order to improve clarity, we restructured our conclusions to summarize our findings in more detail.
  2. In Figure 1 the direction of writing requires the reader to turn the screen, which negatively impacts the function of the figure – providing an easy-access overview of the content. I strongly suggest to provide the writing in a horizontal fashion. The same accounts for Figure 2. We have now updated both figures in order to correct for this problem. We have rotated GPR30 and ERα in both figures by 180 degrees to make the reader experience become easier.
  3. In lines 79/80 I suggest to revise the definition of hypertension according to current clinical guidelines (i.e. AHA or ESC guidelines) and cite accordingly. We have now defined hypertension based on the current AHA guidelines and cited accordingly.
  4. The authors state that hypertension is higher amongst women in individuals aged >60 years. The cited publication ‘16’ reports results specific to Korea. Is there any further data in this respect? Otherwise I suggest that the statement needs to be tuned down, since it seems a bit surprising. We thank the reviewer for their comment. We have now incorporated a study from Centers for Disease Control and Prevention on this matter. This study includes data from the National Health and Nutrition Examination Survey during the years 2015 and 2016.
  5. In line 177, the authors first mention BMPR2. I suggest to provide an introductory sentence to this receptor, which explains the connection to the topic it is used under. We have now added an introductory sentence explaining BMPR2 and its connection to pulmonary artery hypertension.
  6. Please check first mention of PASMC and its initial explanation. We have now made this correction in the text.
  7. Please check authors in reference ‘13’. We have made this correction.

Reviewer 2 Report

Excellent overview of current studies on estrogen, estrogen receptors and their contribution to cardiovascular health and disease! 

However, the reader may get lost among the well cited but numerous details. Thus, may it be possible to sum up the essence of findings and possible clinical implication in a short paragraph at the end of each chapter?

Minor points:

Abstract: line 13: '...a number that is expected to grow rapidly.'

This expectation may be deleted, since it is not that clear because CVD is decreasing in many countries while e.g. infections diseases are getting more common among elderly.

Minor points may be corrected by the publisher's reader, e.g. line 45: ( too many; line 52: has instead of have

In Fig. 1 and 2 the writing of the two fields referring to GPR30 and Era may be turnen by 180° for better legibility.

Author Response

Response to Reviewer #2:

Excellent overview of current studies on estrogen, estrogen receptors and their contribution to cardiovascular health and disease! 

We thank the Reviewer and appreciate the time and effort that have been taken to provide constructive comments on our work.

  1. However, the reader may get lost among the well cited but numerous details. Thus, may it be possible to sum up the essence of findings and possible clinical implication in a short paragraph at the end of each chapter? We thank the reviewer for their comment. We have now incorporated a paragraph at the end of each sub-section to sum up the important findings of that section.
  2. Abstract: line 13: '...a number that is expected to grow rapidly.' This expectation may be deleted, since it is not that clear because CVD is decreasing in many countries while e.g. infections diseases are getting more common among elderly. We thank the reviewer for their comment and have now deleted “…a number that is expected to grow rapidly” from the text.
  3. Minor points may be corrected by the publisher's reader, e.g. line 45: ( too many; line 52: has instead of have. We have now made these corrections in the text.
  4. In Fig. 1 and 2 the writing of the two fields referring to GPR30 and Era may be turnen by 180° for better legibility. We have now updated both figures in order to correct for this problem. We have rotated GPR30 and ERα in both figures by 180 degrees to make the reader experience become easier.

Reviewer 3 Report

The article is good written and the authors made a good review of the literature.

I do have some minor comments, mainly concerning editing:

  • I don't find in the text references 111 and 128
  • line 246: references 60 and 61 should be together in the same brackets
  • line 312: seems to me you have made reference to ref 38 before ref 77 (line 318)
  • Table 3: second line from bottom is different from the others
  • line 357: you make reference to ref 89 but I didn't ref 88 before
  • line 489: need to format refs 112-114. Is it 112 to 114?? Then ref 111 is here after all... (first of my points)
  • line 484: paragraph missing before table 6 title; table 6 title should not be in bold

Author Response

Response to Reviewer #3:

The article is good written and the authors made a good review of the literature.

We thank the Reviewer for their time and constructive comments on our review.

I do have some minor comments, mainly concerning editing:

  1. I don't find in the text references 111 and 128. We thank the Reviewer for their comment. We have updated our reference list and this mistake is now corrected. All references are now mentioned in text.
  1. line 246: references 60 and 61 should be together in the same brackets. We have now made this correction.
  2. line 312: seems to me you have made reference to ref 38 before ref 77 (line 318). We have now corrected for this and the references are now in order.
  3. Table 3: second line from bottom is different from the others. We have now fixed the second line from the bottom in Table 3.
  4. line 357: you make reference to ref 89 but I didn't ref 88 before. This correction has now been made.
  5. line 489: need to format refs 112-114. Is it 112 to 114?? Then ref 111 is here after all... (first of my points). We have now corrected the format of this reference. There are no longer any references that are not listed.
  6. line 484: paragraph missing before table 6 title; table 6 title should not be in bold. We have now corrected the formatting of the paragraph before Table 6. Also, Table 6 title is no longer bolded.